# Molecular Responses in THP-1 Macrophage-Like Cells Exposed to Diverse Nanoparticles

**DOI:** 10.3390/nano9050687

**Published:** 2019-05-02

**Authors:** Tana Brzicova, Eliska Javorkova, Kristyna Vrbova, Alena Zajicova, Vladimir Holan, Dominik Pinkas, Vlada Philimonenko, Jitka Sikorova, Jiri Klema, Jan Topinka, Pavel Rossner

**Affiliations:** 1Department of Genetic Toxicology and Nanotoxicology, Institute of Experimental Medicine of the Czech Academy of Sciences, 14220 Prague, Czech Republic; tana.brzicova@iem.cas.cz (T.B.); kristyna.vrbova@iem.cas.cz (K.V.); jitka.sikorova@iem.cas.cz (J.S.); jan.topinka@iem.cas.cz (J.T.); 2Laboratory for Risk Research and Management, Faculty of Safety Engineering, VSB—Technical University of Ostrava, 70030 Ostrava, Czech Republic; 3Department of Transplantation Immunology, Institute of Experimental Medicine of the Czech Academy of Sciences, 14220 Prague, Czech Republic; eliska.javorkova@iem.cas.cz (E.J.); alena.zajicova@iem.cas.cz (A.Z.); vladimir.holan@iem.cas.cz (V.H.); 4Microscopy Centre, Institute of Molecular Genetics of the Czech Academy of Sciences, 14220 Prague, Czech Republic; dominik.pinkas@img.cas.cz (D.P.); vlada.philimonenko@img.cas.cz (V.P.); 5Institute for Environmental Studies, Faculty of Science, Charles University, 12801 Prague, Czech Republic; 6Department of Computer Science, Czech Technical University in Prague, 12135 Prague, Czech Republic; klema@fel.cvut.cz

**Keywords:** nanoparticles, THP-1 cells, immune response, reactive oxygen species, DNA damage

## Abstract

In the body, engineered nanoparticles (NPs) may be recognized and processed by immune cells, among which macrophages play a crucial role. We evaluated the effects of selected NPs [NM-100 (TiO_2_), NM-110 (ZnO), NM-200 (SiO_2_), and NM-300 K (Ag)] on THP-1 macrophage-like cells. The cells were exposed to subcytotoxic concentrations of NPs (1–25 µg/mL) and the expression of immunologically relevant genes (VCAM1, TNFA, CXCL8, ICAM1, CD86, CD192, and IL1B) was analyzed by RT-qPCR. The expression of selected cytokines, growth factors and surface molecules was assessed by flow cytometry or ELISA. Generation of reactive oxygen species and induction of DNA breaks were also analyzed. Exposure to diverse NPs caused substantially different molecular responses. No significant effects were detected for NM-100 treatment. NM-200 induced production of IL-8, a potent attractor and activator of neutrophils, growth factors (VEGF and IGF-1) and superoxide. NM-110 triggered a proinflammatory response, characterized by the activation of transcription factor NF-κB, an enhanced production of proinflammatory cytokines (TNF-α) and chemokines (IL-8). Furthermore, the expression of cell adhesion molecules VCAM-1 and ICAM-1 and hepatocyte growth factor (HGF), as well as superoxide production and DNA breaks, were affected. NM-300 K enhanced IL-8 production and induced DNA breaks, however, it decreased the expression of chemokine receptor (CCR2) and CD86 molecule, indicating potential immunosuppressive activity. The toxicity of ZnO and Ag NPs was probably caused by their intracellular dissolution, as indicated by transmission electron microscopy imaging. The observed effects in macrophages might further influence both innate and adaptive immune responses by promoting neutrophil recruitment via IL-8 release and enhancing the adhesion and stimulation of T cells by VCAM-1 and ICAM-1 expression.

## 1. Introduction

From the perspective of the immune system, engineered nanomaterials (NMs) represent foreign objects in the body. As such they may be recognized and processed by immunocompetent cells, especially those involved in the innate immune response providing a rapid, nonspecific response to potential threats. Macrophages, as professional phagocytes, represent a crucial cell component of innate immunity. Macrophages act as the first line of defense against invading agents, including NMs. In vivo studies have shown that resident lung, liver, or spleen macrophages are involved in the removal of (nano)particles [1]. Interactions of nanomaterials (NMs) with macrophages may determine not only the fate of NMs (e.g., by NM clearance) but also their toxic effects (e.g., by triggering inflammation and reactive oxygen species production). Macrophages play an important role in the pathogenesis of various diseases caused by exposure to (nano)particles and (nano)fibers, such as lung fibrosis or mesothelioma [2]. THP-1—the human monocytic leukemia cell line—can be differentiated into adherent macrophage-like cells by phorbol-12-myristate-13-acetate (PMA) [3]. The differentiated THP-1 cells have been shown to represent a suitable model system for studying macrophage functions in vitro [4].

The biological effects of NMs are determined by a combination of various properties including their shape, size, surface properties, etc. The chemical composition, especially the solubility and toxicity of the parent substance, is still considered to be one of the main factors affecting the interactions of NMs with biological systems. The chemical composition is also among the key characteristics for NM categorization for regulatory purposes [5].

In this study, we used four NM samples of different chemical composition, obtained from the Joint Research Centre (JRC) Nanomaterials Repository. We compared the effects of soluble ZnO and Ag nanoparticles (NPs) (NM-110 and NM-300 K, respectively), and nonsoluble TiO_2_ and SiO_2_ NPs (NM-100 and NM-200, respectively). To get an insight into immunologically mediated events triggered by NMs in macrophages, we analyzed the changes of expression of selected genes and their protein products in THP-1-derived macrophage-like cells exposed to the above-mentioned NMs. We focused here on VCAM1, TNFA, CXCL8, ICAM1, CD86, CD192 and IL1B encoding VCAM-1 (CD106), TNF-α, IL-8, ICAM-1 (CD54), CD86, CCR2 (CD192) and IL-1β proteins, respectively. These proteins are responsible for cell adhesion (VCAM-1, CCR2) and the extravasation of leukocytes (ICAM-1), as well as the activation of T cells resulting in cytokines production (CD86). TNF-α and IL-1β induce the expression of VCAM-1 which is a chemokine inducing cell migration. Additionally, we assessed the expression of proteins with proinflammatory and anti-inflammatory activity (IL-6 and IL-10, respectively) and production of growth factors, including LIF, IGF-1, HGF, VEGF, KGF/FGF-7, FGF basic, and β-NGF. As toxic effects of NPs are often associated with oxidative stress induction resulting in DNA damage, we also focused on reactive oxygen species (ROS) generation and formation of DNA breaks. On the whole, whereas TiO_2_ NPs did not exhibit significant effects in our experimental system, other NPs (NM-110, NM-200, NM-300 K) induced a dose-dependent immunomodulatory response. Oxidative stress-related response was weak, mostly limited to exposure to NM-110 and NM-200. Our data might be useful in elucidating NP-specific modes of action and mechanisms of potential toxicity.

## 2. Materials and Methods

### 2.1. Nanomaterials

Four different types of NPs [NM-100 (TiO_2_), NM-110 (ZnO), NM-200 (SiO_2_), and NM-300 K (Ag)] were obtained from the JRC nanomaterials repository. 

Samples of NM-100, NM-110, and NM-200 were supplied in the form of dry powders, NM-300 K was a colloidal dispersion in deionized water (85%) containing 7% stabilizing agent (ammonium nitrate) and 8% emulsifiers (4% Polyoxyethylene Glycerol Trioleate and 4% Polyoxyethylene Sorbitan Monolaurate (Tween 20) with a nominal silver concentration of 10% (*w*/*w*)). 

NM stock dispersions (2.56 mg/mL) were prepared according to the SOP developed within the NANOGENOTOX project [6]. Briefly, NMs were prewetted with 96% EtOH (0.5 vol%) to enable the dispersion of hydrophobic NMs in water-based systems. Sterile-filtered water containing 0.05% *w*/*v* BSA (99.5 vol%) was added as a dispersion medium. The stock dispersion at a concentration of 2.56 mg/mL was sonicated in a scintillation vial placed in an ice-water bath using 400 W, 20 kHz Branson Sonifier S-450 D (Branson Ultrasonics Corp.; Danbury, CT, USA) equipped with a 13 mm disruptor horn. The 16 min sonication at 10% amplitude delivered acoustic energy of 7056 J. Sonicated dispersions were diluted in a cell culture medium to the required concentrations immediately before exposure. NMs dispersions were vortexed for 10 s before handling to ensure their homogeneity. Only freshly prepared NM dispersions were used for the experiments.

### 2.2. Hydrodynamic Size and Zeta Potential Measurements

The particle size distribution and zeta potential of NMs in dispersions were measured by dynamic light scattering (DLS) using ZetaSizer Nano ZS (Malvern Instruments Ltd.; Malvern, UK). To control the stability of NM dispersions in complete cell culture media containing 10% FBS HI (CCM), the measurements were performed in dispersions stored in a CO_2_ incubator at 37 °C for 1 and 24 h. The zeta potential was calculated from the electrophoretic mobility using the Helmholtz–Smoluchowski equation.

### 2.3. Cell Cultivation, Differentiation and Exposure

The THP-1 cell line (human monocytic leukemia cells; ATCC, Manassas, VA, USA) was cultured in an RPMI 1640 GlutaMAX medium (Gibco, Waltham, MA, USA) supplemented with 10% (*v*/*v*) heat-inactivated fetal bovine serum (FBS HI; Sigma-Aldrich, Saint Louis, MO, USA). Cells were cultured at a density ranging from 200,000 to 800,000 cells per mL in a humidified incubator at 37 °C and 5% CO_2_. 

For the experiments, the cells were differentiated into macrophage-like cells by cultivation in CCM supplemented with 100 nM PMA for 72 h. The PMA-containing medium was then removed and the cells were washed with CCM and incubated for 24 h in complete media without PMA (resting period).

Cytotoxicity assays were performed on THP-1 cells, seeded into 96-well plates at a density of 5 × 10^4^ cells per well in 100 µL CCM. ROS production was evaluated in 96-well black wall clear bottom plates. For the gene expression analyses and flow cytometry, the cells were seeded into 60 mm Petri dishes at a concentration of 3.15 × 10^6^ cells per dish in 6.3 mL CCM. The cells for the comet assay were exposed in 6-well plates at a density of 1.45 × 10^5^ cells per well in 3 mL CCM. For TEM imaging, THP-1 cells were seeded, differentiated and exposed onto 12 mm glass coverslips (Schott Glass AG, Mainz, Germany) placed in wells of a 24-well plate (3 × 10^5^ cells per well in 0.6 mL CCM). The cell number and concentrations of NPs per surface area were kept comparable across all experiments. Tested concentrations of NPs were obtained by serial dilutions in CCM, supplemented with 100 U/mL penicillin, 100 mg/mL streptomycin, and containing 1% milliQ water with 0.05% BSA (*v*/*v*) to ensure an identical vehicle in each well. Prior to exposure, CCM was removed, the cells were washed with CCM and treated with freshly prepared dilutions of NMs, media only (negative controls; NC) or to the equivalent weight of dispersant present in the NM-300 K sample. Lipopolysaccharide (LPS; Sigma-Aldrich, St. Louis, MO, USA) at a concentration of 10 µg/mL was used as a positive control (PC). 

### 2.4. MTS Assay

Cellular viability was determined by the MTS assay that evaluates the reduction of 3-(4,5-dimethylthiazol-2-yl)-5-(3-carboxymethoxyphenyl)-2-(4-sulfophenyl)-2H-tetrazolium (MTS) to formazan by enzymatic activity (mitochondrial dehydrogenases) of viable cells. THP-1 cells were seeded and differentiated as described above. After 24-h exposure, the cells were washed twice with a phosphate-buffered saline (PBS), and 120 µL MTS reagent (CellTiter 96 Aqueous One Solution Cell Proliferation Assay; Promega, Madison, WI, USA) diluted in phenol red free RPMI medium at a ratio of 1:5, which was added to each well. After 1-h incubation in the dark at 37 °C with a 5% CO_2_ atmosphere, 100 µl of the MTS containing media was transferred to the appropriate wells of a new plate to avoid interference of the NMs adsorbed on the plate surface with the absorbance read-out. The formazan absorbance was measured at 490 nm using a SpectraMax^®^ M5 Plate Reader (Molecular Devices, Sunnyvale, CA, USA). To determine the viability, the background values of the wells without cells were subtracted from the values of the tested wells. The mean absorbance of the untreated cells served as the reference for calculating 100% cellular viability. The results were expressed as a percentage of the viability of the untreated cells. 

Background absorbance measurements of NPs in CCM and the MTS reagent without cells were performed to examine the potential interference of NPs with the MTS assay. 

### 2.5. RNA Isolation and Quality Control

Total RNA from lysed THP-1 macrophage-like cells was obtained using NucleoSpin RNA II according to the manufacturer’s instructions. RNA concentration was quantified with a Nanodrop ND-1000 Spectrophotometer (Thermo Fisher Scientific, Waltham, MA, USA), and the quality was checked on an Agilent 2100 Bioanalyzer (Agilent Technologies Inc.; Santa Clara, CA, USA). All samples had an RNA Integrity Number (RIN) higher than 9. Isolated RNA was stored at −80 °C until use.

### 2.6. Real-Time Quantitative PCR (RT-qPCR)

One µg of RNA from each sample was reverse transcribed to generate complementary DNA (cDNA) using the Transcriptor High Fidelity cDNA Synthesis Kit (Roche, Mannheim, Germany). The original protocol was modified by using 2.5 μM oligo(dT) and 10 μM random hexamers for priming. Samples with primers were denatured 10 min at 65 °C and then put on ice. cDNA synthesis was performed using 1x THF reaction buffer, 20 U Protector RNase inhibitor, 1 mM dNTP mix, 5 mM DTT, 10 U THF Reverse transcriptase according to the following condition; 10 min at 25 °C, 30 min at 55 °C, and 5 min at 85 °C. Quantitative PCR measurements were performed using the LightCycler 480 (Roche, Basel, Switzerland). Each RT-qPCR reaction was carried out in a final volume of 14 μL containing 2.5 μL of diluted cDNA (25 ng cDNA in reaction mix), 3.8 μL of water and 7 μL of master mix. To determine the level of each target gene, 0.7 μL of a specifically designed assay (Custom designed real-time PCR assay with Double-Dye probe, Primerdesign; Southampton, UK) was added to the reaction mixture. The cycling conditions were: initial DNA release and denaturation at 95 °C for 2 min, followed by 40 cycles of amplification (10 s at 95 °C and 60 s at 60 °C). Ct values were generated by GenEx software (MultiD AB). The expression levels of target genes were normalized to the reference genes (ATP5B and ACTB). The reference genes were selected according to the stability of gene expression during experimental conditions using the geNorm Reference Gene Selection Kit. The sequences of the primers used in RT-qPCR are shown in Table 1.

### 2.7. ELISA

The release of selected cytokines and growth factors (IL-1β, IL-6, IL-8, IL-10, TNF-α, LIF, IGF-I, HGF, VEGF, KGF/FGF-7, FGF basic, and β-NGF) into CCM was quantified by an enzyme-linked immunosorbent assay (ELISA). Supernatants from cultures containing 0.5 × 10^6^ cells/mL in a final volume of 1 mL CCM were harvested after 24-h exposure period and stored at −80 °C until analyses. DuoSet ELISA kits for the detection of all above cytokines were purchased from R&D Systems (Minneapolis, MN, USA). The reaction was quantified by spectrophotometry using a Sunrise Remote ELISA Reader (Tecan Ltd.; Männedorf, Switzerland). Cytokine concentrations were determined by interpolation from the standard curve.

To take into account possible interferences, the cytokine-binding capacity of the tested NPs was evaluated by adding NPs at the highest tested concentrations to supernatants harvested from PC (cells stimulated with 10 µg/mL of LPS) and incubated at 37 °C, 5 % CO_2_ for 24 h. 

### 2.8. Flow Cytometry

The expression of cell surface molecules was analyzed by flow cytometry. Scraped cells were incubated for 30 min at 4 °C with the following monoclonal antibodies (mAbs) (all purchased from BioLegend, San Diego, CA, USA); allophycocyanin (APC)-labeled anti-CD54 (clone HA58), phycoerythrin (PE)-labeled anti-CD86 (clone IT2.2), PE-labeled anti-CD106 (clone 429), and APC-labeled anti-CD192 (clone K036C2). The potential interference of NPs with binding of mAbs was examined in NP dispersions incubated with the particular mAb. Dead cells were stained with Hoechst 33,258 dye (Sigma Aldrich, St. Louis, MO, USA). Data were collected using an LSRII flow cytometer (BD Bioscience, Franklin Lakes, NJ, USA).

### 2.9. ROS Production

Generation of superoxide and other reactive oxidative species (including hydrogen peroxide, peroxynitrite, hydroxyl radicals, nitric oxide, and peroxy radical) was quantified using a Cellular ROS/Superoxide Detection Assay (Abcam, Cambridge, UK) according to the manufacturer’s instructions. Tert-butyl hydroperoxide was used as a positive control. Fluorescence was measured using SpectraMax^®^ M5 Plate Reader (Molecular Device, Sunnyvale, CA, USA) at Ex = 488 nm, Em = 520 nm and Ex = 550 nm, Em = 610 nm for the detection of ROS and superoxide, respectively. Results were expressed as relative fluorescence intensity normalized to the fluorescence of the negative control (untreated cells).

### 2.10. Comet Assay

DNA strand breaks were evaluated by the alkaline version of the comet assay according to Sigh et al. [7] with some modifications. DNA base excision repair enzyme formamidopyrimidine-DNA glycosylase (FPG) was employed to detect oxidized purines. After 24-h exposure, cells were detached by trypsinization, centrifuged, resuspended in cold PBS, and embedded in 0.7% low melting point agarose on a microscope slide precoated with normal melting agarose covered. Cells were lysed with 100 mM EDTA, 2.5 M NaCl, 10 mM EDTA, and 1% Triton X-100 (pH 10.0) for 20 h at 4 °C. After lysis, the samples were washed three times for 5 min in FPG buffer (40 mM HEPES, 0.1 M KCl, 0.5 mM EDTA, 0.2 mg/mL BSA) and incubated with buffer or buffer containing FPG (New England Biolabs, UK) for 30 min at 37 °C. After washing, slides were immersed in unwinding solution (300 mM NaOH, 1 mM EDTA, pH > 13) for 30 min at 4 °C. Electrophoresis was performed in fresh unwinding solution for 30 min at 4 °C (1.2 V/cm, 300 mA). Slides were rinsed twice with neutralization buffer (0.4 M Tris, pH 7.5) for 5 min. DNA was stained with ethidium bromide (20 mg/mL). Comet images were captured with a CCD-1300B camera (VDS Vosskuhler, Germany) attached to a BX51 fluorescence microscope (Olympus, Japan) and analyzed using Lucia Comet Assay 7.00 software (Laboratory Imaging, Prague, Czech Republic). The total DNA damage was measured in 100 randomly selected cells per slide. Results were expressed as a percentage of DNA in the tail (tail DNA %). Sites sensitive to FPG (referred to as FPG sites) were obtained by subtracting values of FPG non-treated slides (Buffer control) from FPG treated slides. Cells treated with 1 mM CdSO_4_ for 2 h were used as a positive control.

### 2.11. Transmission Electron Microscopy

NPs internalization by THP-1 macrophage-like cells was verified by transmission electron microscopy (TEM). After 24-h incubation with 25 µg/mL NPs, cells were quickly washed with Sörensen buffer (0.1 M sodium/potassium phosphate buffer, pH 7.3; SB) at 37 °C, fixed with 2.5% glutaraldehyde in SB for 2 h, washed with SB and postfixed with 1% OsO_4_ solution in SB for 2 h. The cells were dehydrated in a series of acetone with an increasing concentration and embedded in Epon–Durcupan resin. After polymerization for 72 h at 60 °C, blocks were cut into 80 nm ultrathin sections and collected on 200 mesh size copper grids. The sections were examined in an FEI Morgagni 268 transmission electron microscope operated at 80 kV. The images were captured using a Mega View III CCD camera (Olympus Soft Imaging Solutions).

### 2.12. Statistical Analysis

The baseline and threshold values of RT-qPCR experiments raw data were assessed with GenEx software version 6.1 (MultiDAnalyses AB) for determination of Ct values. The expression levels of target genes were normalized to the reference genes (ATP5B and ACTB). The reference genes were selected according to the stability of gene expression during experimental conditions using the geNorm Reference Gene Selection Kit (Primerdesign, Southampton, UK). The gene expression levels were compared with those in the untreated control cells. The relative changes in the normalized gene levels were calculated using the 2^−ΔΔCt^ method [8]. Multiple testing correction was performed using the method of Benjamini et al. [9].

In flow cytometry experiments, 20,000 events from each sample after the exclusion of cell debris, dead cells, and cell clusters were analyzed using FlowJo software (LLC, Ashland, Covington, KY, USA). Fluorescent positivity was gated based on the appropriate FMO controls.

The Prism 5 program (GraphPad Software, San Diego, CA, USA) was used for the statistical analysis of the MTS assay, flow cytometry, ELISA, ROS production, and the comet assay results. All experiments were repeated at least three times. In the comet assay, the median values of 100 randomly selected cells from three independent experiments were used for statistical analysis. Data are expressed as the mean ± standard deviation (SD). The statistical significance of differences between the means of individual groups was calculated using a one-way analysis of variance (ANOVA) with Dunnett’s post hoc test. The differences were considered significant for a p-value less than 0.05. IC50 values (concentrations that inhibited cell survival by 50%) for the MTS assay were calculated using four-parameter log-logistic models in the drc package in the statistical software R (version 3.4.0.) [10].

## 3. Results

### 3.1. NM Characterization

Detailed characterization of the NMs is provided in the corresponding JRC reports and publications [11,12,13,14]. Key NP characteristics are reported in Table 2.

Particle size distribution in dispersions and zeta potential representing the surface charge of NPs were analyzed using DLS. To assess the stability of the dispersions, measurements were performed in freshly sonicated samples in ddH_2_O-BSA as well as diluted in CCM at the beginning (1 h) and the end (24 h) of the incubation period. A hydrodynamic size of NM-100 and NM-300 K was similar in particles dispersed in ddH_2_O-BSA and diluted in CCM, even after 24-h incubation, suggesting the formation of relatively stable aggregates and agglomerates. The average values of hydrodynamic size of NM-110 and NM-200 decreased after dilution in CCM. The zeta potential of ddH_2_O-BSA dispersions ranged from −4.8 mV for NM-300 K to −32.9 mV for NM-200. The opacity of NM-100 did not allow the measurement of zeta potential of the concentrated sample. After dilution in CCM, however, the zeta potential of all the samples had similar values, around −16 mV, that did not change significantly after 24-h incubation (Table 3). Negative zeta potential values indicate the adsorption of BSA which carries a negative charge at physiological pH [15]. BSA was used as a dispersant in NP water suspensions and is also the main component of the FBS added to CCM.

### 3.2. Cell Viability Measured by MTS

The results presented in Figure 1 show that NM-100 did not affect cell viability up to the highest tested concentration of 100 µg/mL. Other samples caused a significant decrease in cell viability when compared to untreated cells at concentrations of 25 µg/mL for NM-300 K and 50 µg/mL for both NM-110 and NM-200. Based on the dose-response curves, noncytotoxic concentrations (1, 10 and 25 µg/mL) were selected for gene expression and DNA damage experiments with NM-100, NM-110, and NM-200. Due to the higher cytotoxic potential of NM-300 K (51% viability at 25 µg/mL), an additional concentration of 15 µg/mL was used for this sample. For ELISA and flow cytometry experiments, an additional concentration was also used for the other NPs: 50 µg/mL corresponding to 64% and 66% viability in NM-110 and NM-200, respectively, and 100 µg/mL for noncytotoxic NM-100.

### 3.3. Transmission Electron Microscopy

Cellular uptake and intracellular morphology upon NP exposure (25 µg/mL) were investigated in ultrathin sections of resin-embedded cells using transmission electron microscopy (TEM). TEM is commonly used for NPs characterization, although possible damage of specimen by high-voltage electron beams has been reported [16]. This fact should not affect our results because in this study TEM was mainly used to detect the presence NPs in the cells. As shown in Figure 2A,C, TEM imaging confirmed internalization of NM-100 and NM-200 by THP-1 macrophage-like cells. The prevalent localization of NPs in a form of aggregates and agglomerates in cell vesicles, such as phagosomes, lysosomes, and endosomes (Figure 2A’,C’) suggests that particles were internalized through an active process (probably endocytosis). NM-110 was not observable inside the exposed cells (Figure 2B,B’), which may indicate the dissolution of ZnO NPs into Zn^2+^ under endosomal acidic pH. NM-300 K was detected both as individual NPs and NP clusters in cell vesicles (Figure 2D). The structures shown in Figure 2D’ might form as a result of a possible binding of dissolved silver ions on intracellular molecules (e.g., proteins).

### 3.4. mRNA Expression of Immunologically Relevant Genes

We first analyzed the effects of NP treatment on mRNA expression of selected immunologically relevant genes. These genes were selected based on the results of whole-genome gene expression analysis that showed a deregulation of a number of immune response- and inflammation-related transcripts (manuscript under preparation). The selected genes and corresponding proteins are reported in Table 4.

The treatment with NM-100 had no effect on mRNA expression of IL1B, CXCL8 and TNFA genes; all the other tested NMs increased the expression of these genes in a dose-dependent manner (Figure 3A–C). A particularly strong effect was observed for these genes upon exposure to NM-110 (25 µg/mL, Figure 3A–C). In general, for TNFA, expression changes were most pronounced for CXCL8, and lowest, but still significant, at higher concentrations (Figure 3B,C).

A strong effect of NM-110 exposure on upregulation of VCAM1 expression was detected, but only after the treatment with the highest concentration of this NM (Figure 3D). We observed a dose-dependent upregulation of ICAM1 expression after exposure to all of the tested NPs. The changes were statistically significant at higher concentrations (25 μg/mL in NM-100 and 10 μg/mL in other NPs) (Figure 3E). We also observed weak effects of NPs exposure on CD86 mRNA expression: the lowest concentration of NM-100 increased the expression and the highest concentrations of NM-300 K decreased the expression of the gene (Figure 3F). The expression of CD192 mRNA was downregulated in a dose-dependent manner after treatment of the cells with NM-110 and NM-300 K (Figure 3G). 

### 3.5. Surface Molecule Expression Assessed by Flow Cytometry

The mRNA expression results for selected surface molecules were compared with the respective protein levels using flow cytometry. The increased expression of ICAM-1 and VCAM-1 in cells exposed to NM-110 corresponds with RT-qPCR results (Figure 4A,B). Interestingly, increased VCAM-1 expression was detected in lower concentrations using flow cytometry rather than upregulation of the corresponding gene in RT-qPCR. However, in contrast to RT-qPCR, the flow cytometry analysis did not detect an increased ICAM-1 expression in cells treated with other NMs (NM-200 and NM-300 K) (Figure 4A). The decrease in expression of CD86 upon exposure to NM-300 K, as well as expression of CCR2 after both NM-110 and NM-300 K treatment, was in agreement with the downregulation of the respective mRNA levels in RT-qPCR, although mRNA levels were decreased in lower concentrations rather than corresponding molecules (Figure 4C,D). Changes in surface molecule expression were dose-dependent, with the effects also observed in cytotoxic concentrations (25 μg/mL for NM-300 K and 50 μg/mL for NM-110 and NM-200). In contrast to RT-qPCR, flow cytometry detected no significant changes in the selected surface markers in cells exposed to NM-100.

### 3.6. Cytokine Production Measured by ELISA

We then performed ELISA to compare RT-qPCR data with the protein levels of the selected cytokines IL-1β and TNF-α and chemokine IL-8, and for additional cytokines and growth factors IL-6, IL-10, LIF, IGF-I, HGF, VEGF, KGF/FGF-7, FGF basic, and β-NGF, which are produced by THP-1 cell line. Similar to mRNA expression, the concentrations of IL-8 in CCM from cells exposed to NM-110, NM-200, and NM-300 K exhibited the dose-dependent increase, reaching the values of LPS, a potent activator of macrophages (Figure 5B). In concordance with the RT-qPCR data, the concentration of 1 µg/mL of all NPs did not have any effect on the chemokine production in the exposed cells. NM-100 did not induce IL-8 production at any of the tested concentrations (Figure 5B). TNF-α production was increased in NM-110-exposed cells, reaching statistical significance at 50 µg/mL (Figure 5C). This is in contrast with mRNA results, where a significant increase of mRNA expression was observed at lower concentrations (10 µg/mL) of all but one (NM-100) tested NMs. Interestingly, concentrations of IL-1β remained at low, nonsignificant levels (Figure 5A), contrarily to RT-qPCR data that reported a significant IL1B gene upregulation in NM-110, NM-200 and NM-300 K exposed cells. All other cytokines and growth factors (except for HGF, IGF-1, and VEGF) detected on protein level by ELISA were produced at low quantities and their production was not increased by the presence of NPs. A significant increase in the secretion of HGF, IGF-1 and VEGF was recorded after incubation with of NM-110 and NM-200 NPs, respectively (Figure 5D–F). For HGF, a significant decrease of protein levels was further detected after exposure to higher doses (25 a 50 µg/mL) of NM-110.

### 3.7. Generation of Reactive Oxygen Species and DNA Damage

We analyzed ROS generation and separately superoxide production after exposure to the tested NMs. Overall, the effects on ROS levels were weak and the levels did not significantly increase even after 24-h incubation. An inconsistent significant decrease was observed after NM-300 K exposure (Figure 6A). NM-110 and NM-200 slightly affected superoxide levels, although the change was mostly limited to longer treatment periods and, for NM-200, no dose-dependent response was observed (Figure 6B).

Similar to the ROS production, DNA damage induced by the studied NMs was weak, even though we observed a nonsignificant dose-dependent increase of DNA strand breaks after treatment with NM-100, NM-110, and NM-200. NM-110 and NM-300 K (25 µg/mL) significantly affected formation of FPG-sensitive sites (Figure 7).

## 4. Discussion

Macrophages represent a crucial cell component of the innate immune system and the first line of defense against potentially dangerous factors invading the body. Occupying practically all tissues and organs, macrophages are among the first immune cells that NPs encounter once they have entered the body. It has been shown that even in the absence of direct toxicity, NPs may interfere with the development of defensive responses which may lead to inadequate immune reactions and eventual pathological consequences.

In this study, we compared the effects of four NMs of a different chemical composition and size [NM-100 (TiO_2_, 190 nm), NM-110 (ZnO, 150 nm), NM-200 (SiO_2_, 50 nm), and NM-300 K (Ag, 20 nm)] in PMA-differentiated THP-1 macrophage-like cells. All chosen NMs are included in the OECD priority list. These NMs are “representative test materials” provided by European Commission’s JRC Nanomaterials Repository. They represent thoroughly characterized samples of commercially available NMs with ensured homogeneity and stability of the batches [17]. The concentrations of NMs were based on pilot experiments and tests of cytotoxicity; for further analyses noncytotoxic doses of NMs were used.

The exposure to diverse NPs caused substantially different molecular responses in THP-1 cells. NM-100, representing low-toxic nonsoluble TiO_2_ NPs, did not exhibit the significant effects up to the highest tested concentration. TEM imaging confirmed the uptake of these NPs in macrophages after 24-hour exposure and their localization in endosomes. TiO_2_ NMs are among the most commonly used NMs in toxicological studies. However, there is no clear conclusion on their potential toxicity. It has been shown that the toxicity of TiO_2_ NMs depends on various properties [18], mainly on their size, crystal structure, and shape, as well as on test conditions (e.g., exposure to light may induce TiO_2_ NPs-mediated phototoxicity), the test method employed, and test system.

Amorphous silica (NM-200), another nonsoluble NM, was cytotoxic in THP-1 macrophages at concentrations of 50 µg/mL and higher. The cytotoxic effects of this NM have been reported to be caused by direct contact of macrophages with silica NPs [19]. TEM imaging confirmed an extensive NP uptake by THP-1 macrophages exposed to 25 µg/mL of NM-200. Internalized NPs were localized mostly in membrane-bound cell vesicles in a form of large aggregates and agglomerates. Constantini et al. [19] have previously shown that silica particles can damage internal cell membranes leading to the leakage of endolysosomal material into the cytoplasm and cell death. These effects were observed to be related to the total administered particle surface area across a wide range of particle diameters (7–500 nm). It should however be noted that some forms of silica nanoparticles are biodegradable and have been suggested as delivery systems for healthcare applications [20].

In noncytotoxic concentrations, NM-200 induced the transcription of a number of chemokines genes (manuscript in preparation). ELISA measurements confirmed a significantly increased production of IL-8 in cells exposed to 10 to 50 µg/mL. The levels of IL-8 in supernatants of cells exposed to 25 µg/mL NM-200 were comparable to the production of IL-8 in cells exposed to LPS. IL-8 is a potent attractor and activator of granulocytes, mainly neutrophils. In in vivo experiments, inhalation of silica NPs has been reported to cause an increase and an accumulation of neutrophils in alveolar walls and BAL fluid [21]. Our results indicate that neutrophil influx and accumulation may be caused by increased concentrations of chemokines, such as IL-8, produced by macrophages as a consequence of phagocytosis of silica NPs. Treatment with these NPs was also associated with superoxide production, although the response was weak and mostly nonsignificant.

In vivo short-term and subchronic inhalation studies with amorphous silica NPs demonstrated reversible lung inflammation, granuloma formation, and focal emphysema without progressive lung fibrosis [22,23].

Exposure to NM-200 at a dose of 50 µg/mL increased the production of two growth factors: IGF-1 and VEGF. IGF-1 acts as an activator of the AKT signaling pathway and an inhibitor of apoptosis [24]. VEGF is an important factor in vasculogenesis and angiogenesis [25]. The role of vascular endothelial growth factor is in angiogenesis [26]. Both growth factors have been implicated in cancer development and progression [27,28]. 

ZnO NPs (NM-110) significantly decreased the viability of macrophages at a concentration of 50 ug/mL. The cytotoxic effects of ZnO NPs are mainly ascribed to released Zn ions. Nano-ZnO represents soluble materials, potentially acting via a Trojan Horse-like mechanism. NPs serve as vectors, enter cells, and release toxic ions in the acidic intracellular environment (in lysosomes or phagolysosomes) [29]. In accordance with the hypothesis of intracellular dissolution, ZnO NPs were not visible inside cells using TEM after 24-h incubation.

It has been proposed that increased intracellular levels of ZnO cause damage to cell compartments (e.g., mitochondria) that are associated with the reactive oxygen species production and induction of oxidative stress. Zinc does not undergo redox cycles between ions of different valency. Therefore, oxidative stress associated with ZnO NPs is probably attributed to cell perturbations leading to an imbalance in cellular redox cycles and ROS production [30]. This is in agreement with increased superoxide production observed in our study after treatment of macrophages with NM-110. Oxidative stress may cause DNA damage (as suggested by our data) and triggers redox-sensitive signaling pathways, such as the MAP kinases and NF-κB cascades that induce production of proinflammatory cytokines and chemokines [31].

Higher NM-110 concentrations induced the expression of TNF-α, a pleiotropic cytokine with strong proinflammatory and immunomodulatory properties. The binding of TNF-α to its receptors leads to the activation of transcription factors AP-1 and NF-κB which in turn induce the genes involved in chronic and acute inflammatory responses. Among other functions, TNF-α acts as a potent chemoattractant for neutrophils, stimulates phagocytosis in macrophages, and promotes the production of other proinflammatory cytokines, such as IL-6 and IL-8. The enhanced production of TNF-α upon exposure to ZnO NPs has been observed both in in vitro experiments on human and murine macrophages [31] and in vivo in BAL of exposed mice. Importantly, increased levels of TNF-α and IL-8 (together with IL-6) were detected in BAL fluid after zinc oxide welding fume exposure [32]. The authors assume that these cytokines and chemokine are involved in the pathogenesis of metal fume fever. Interestingly, NM-110 enhanced expression of cell adhesion molecules ICAM-1 and particularly VCAM-1. These molecules are cytokine-inducible Ig gene superfamily members that bind leukocyte integrins [33]. The induction of VCAM-1 and ICAM-1 is associated with proinflammatory conditions and has also been shown to be elicited by TNF-α. Although VCAM-1 expression was negligible in the control cells (PMA differentiated THP-1 macrophages), NM-110 exposure caused a dose-dependent increase in VCAM-1 surface expression, reaching 35% at the highest tested concentration (50 µg/mL). Cell adhesion molecules play a crucial role in leukocyte adhesion to the endothelium. Their expression in macrophages and other antigen presenting cells (APCs) is much less studied. It has been suggested that cell adhesion molecules on APCs may support the adhesion of lymphocyte and, thus, play a role in antigen presentation and lymphocyte activation [34].

The expression of HGF was observed after exposure to all, but the lowest dose of NM-110. This growth factor activates the tyrosine kinase signaling cascade after binding to the proto-oncogenic c-Met receptor. It plays a central role in angiogenesis, tumorigenesis and tissue regeneration thus playing a role in many types of cancer [35].

Since Ag NPs ionized easily and aggregated in an ion-rich biocompatible solution such as PBS or CCM, we used NM-300 K, a stable aqueous suspension of Ag NPs dispersed in stabilizing agents containing polyoxyethylene glycerol trioleate and Tween 20. The same volume of the dispersant (NM-300 K-DIS) was used as the control sample in this study. NM-300 K-DIS did not affect the cells, therefore, the effects observed in cells exposed to NM-300 K were attributed to the presence of Ag NPs. NM-300 K exhibited cytotoxic effects at a concentration of 25 µg/mL and higher. Similarly to NM-110, Ag NPs toxicity may also be at least partly ascribed to high local intracellular concentrations of silver ions that were introduced to cells via Trojan Horse-type mechanism [36,37]. Solubility of the stabilized NM-300 K in CCM has been reported to be low [23,36,38]. Internalized silver NPs may, however, dissolve in the acidic environment of (phago)lysosomes. The contribution of silver ions to the silver NP toxicity is not clear and there is most probably a combined effect of particles, ions, and precipitates. TEM microscopy confirmed the presence of NM-300 K inside cells, predominantly localized in membrane-bound vesicles. In addition, we observed structures that might be precipitates of dissolved silver. The amount of NPs inside cells incubated with NM-300 K was clearly lower than that observed in NM-100 and NM-200. In agreement with that, Park et al. [36] observed Ag NPs in the cytosol of activated cells and CCM, with the exception of damaged cells. The authors speculated that ionization of the phagocyted Ag NPs induced cytotoxicity.

We observed some similar effects induced by NM-110 and Ag NPs common mechanisms of action. Both NPs caused significant DNA damage after exposure to a dose of 25 µg/mL. Superoxide production was affected, although changes after NM-300 K exposure did not reach statistical significance. Finally, HGF production was elevated following exposure to both NPs.

NM-300 K increased the production of chemokines, namely, IL-8. Interestingly, the expression of chemokine receptor CCR2 was significantly decreased after exposure to NM-300 K suggesting a potential weakened response of NM-300 K exposed macrophages to chemokine-mediated recruitment of effector immune cells to the site of inflammation. We hypothesize, that macrophages tend to persist at the site loaded with Ag NPs attracting other innate immune cells, especially neutrophils via production of IL-8 and other chemokines.

In addition to chemokine receptors, a decrease in mRNA levels was also detected for TLR-8 (toll-like receptor 8) that plays a fundamental role in pathogen recognition and activation of innate immunity, and IFN-αR2 (Interferon Alpha and Beta Receptor Subunit 2) participating in the defense against viruses. The potential immunosuppressive effects of NM-300 K might further be mediated by a decrease in expression of CD86, a crucial costimulatory molecule for T cell activation and survival.

The reduction in the expression of CD86 and TLR-2 by Ag NPs (stabilized with polyvinylpyrrolidone) at both protein and mRNA levels was also reported by Yilma et al. [39] in macrophages infected with *Chlamydia trachomatis*. The authors concluded that Ag NPs were able to lower interactions between macrophages and *C. trachomatis*, cytokines production by innate immune cells and activation of T cells. Bhol et al. [40] reported that Ag NPs suppressed membrane damage and inflammatory-related responses in a rat model of ulcerative colitis.

Numerous studies have, however, reported the rather proinflammatory and immunostimulatory effects of Ag NPs. Inconsistency may be attributed to differences in cell types, cultivation conditions (e.g., composition of CCM), and tested NPs. Ag NPs size and stabilization significantly modify Ag NP–cell interactions as well as the dissolution and aggregation in CCM. 

Interestingly, concentrations of IL-1β were not affected in any of the tested NPs contrarily to gene expression data, indicating that a precursor pro-IL-1β was not further processed into the active protein form.

## 5. Conclusions

In this study, we analyzed the effect of exposure to different types of NMs on the expression of selected immunologically relevant genes and proteins, ROS production and DNA damage. Overall, our results suggest immunostimulating and cell adhesion-promoting effects of NPs on THP-1 macrophage-like cells. More pronounced effects were observed in soluble NPs (zinc oxide NM-110 and silver NM-300 K) suggesting the role of released ions in NP toxicity, mediated, at least in part, by ROS production. We hypothesize that the observed effects in macrophages might further influence both innate and adaptive immune responses by promoting neutrophil recruitment via IL-8 release and enhancing adhesion and stimulation of T cells by VCAM-1 and ICAM-1 membrane expression. Our conclusions, however, need to be interpreted with caution. The THP-1 macrophage-like model, although commonly used as an experimental system, differs in some respects (e.g., phagocytic capacity) from macrophages. These differences may impact the results and their interpretation in context of human organism.

## Figures and Tables

**Figure 1 nanomaterials-09-00687-f001:**
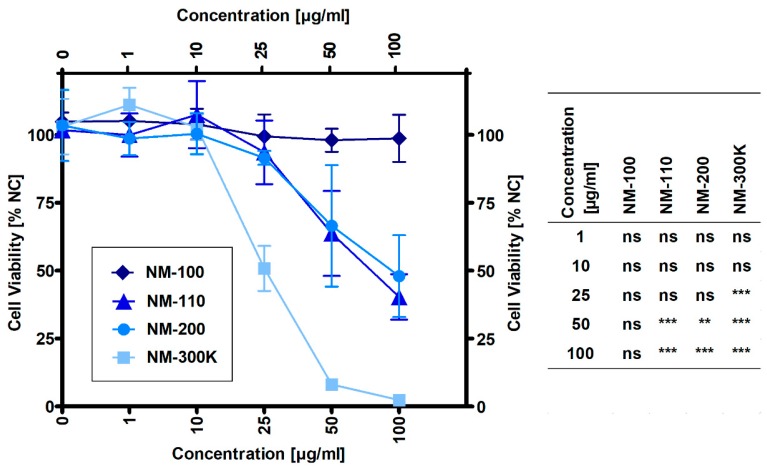
Viability of THP-1 macrophage-like cells exposed to NMs for 24 h evaluated by the MTS assay. Viability was normalized to untreated cells (NC). Results are expressed as mean percentage of cell viability ± SD from three independent experiments. Data were analyzed by ANOVA followed by Dunnett’s Multiple Comparison. ns—nonsignificant, ** *p* < 0.01, *** *p* < 0.001.

**Figure 2 nanomaterials-09-00687-f002:**
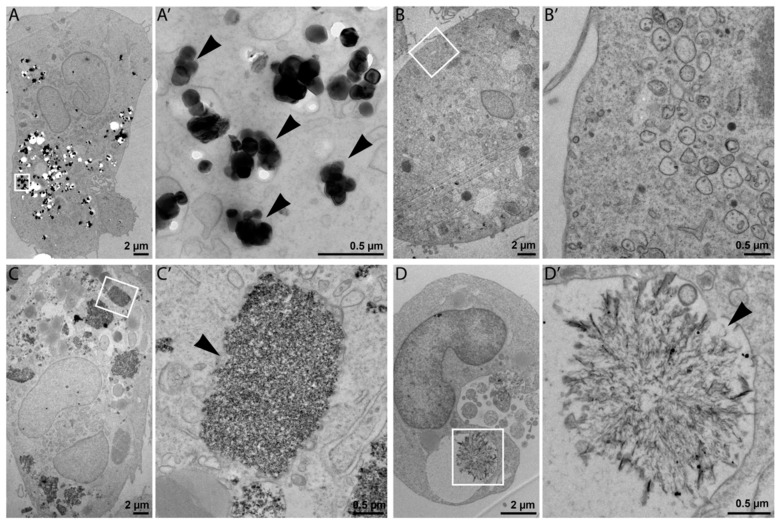
Representative TEM images of THP-1 cells incubated with 25 μg of NPs for 24 h: NM-100 (**A**,**A’**); NM-110 (**B**,**B’**); NM-200 (**C**,**C’**), and NM-300 K (**D**,**D’**). For each type of NP in the pair of images, the left image represents the view of a whole cell (**A**–**D**), and the image to the right represents high magnification detail (**A’**–**D’**) delineated by a white rectangle in the respective low magnification image. Bars, 2 µm (**A**–**D**); 0.5 µm (**A’**–**D’**).

**Figure 3 nanomaterials-09-00687-f003:**
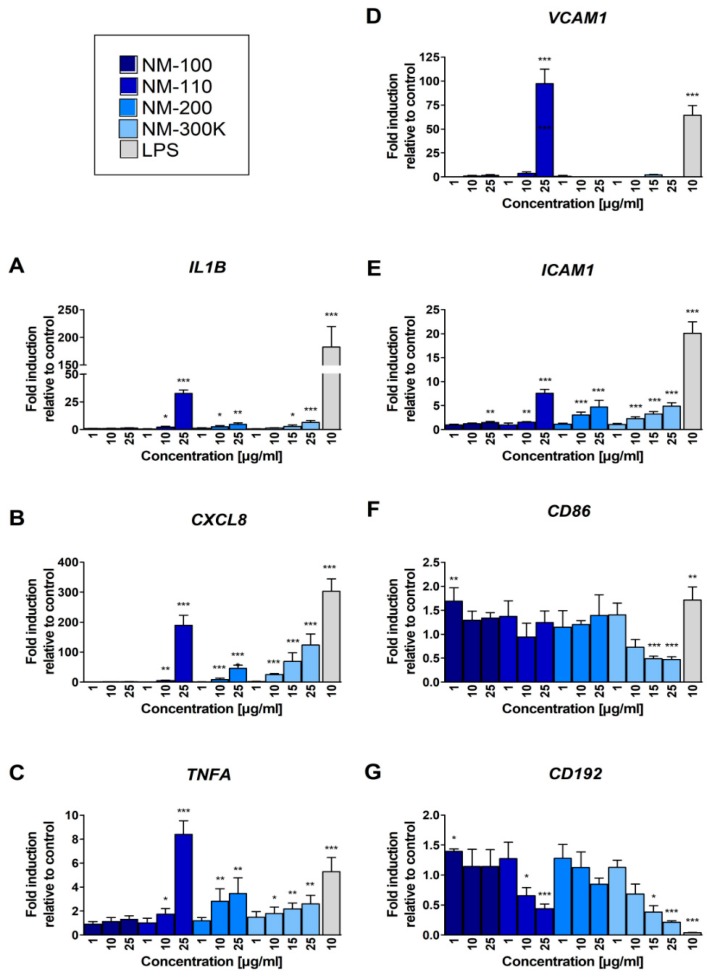
Real-time PCR quantification of fold changes in transcripts of selected immunologically relevant genes after 24-h exposure: IL1B (**A**), CXCL8 (**B**), TNFA (**C**), ICAM1 (**E**), VCAM1 (**D**), CD192 (**G**), and CD86 (**F**). The levels of mRNA were normalized to the untreated cells. Bars represent the mean of three replicates ± SD. Student’s *t*-test was used for statistical analysis of the results. * *p* < 0.05, ** *p* < 0.01, *** *p* < 0.001.

**Figure 4 nanomaterials-09-00687-f004:**
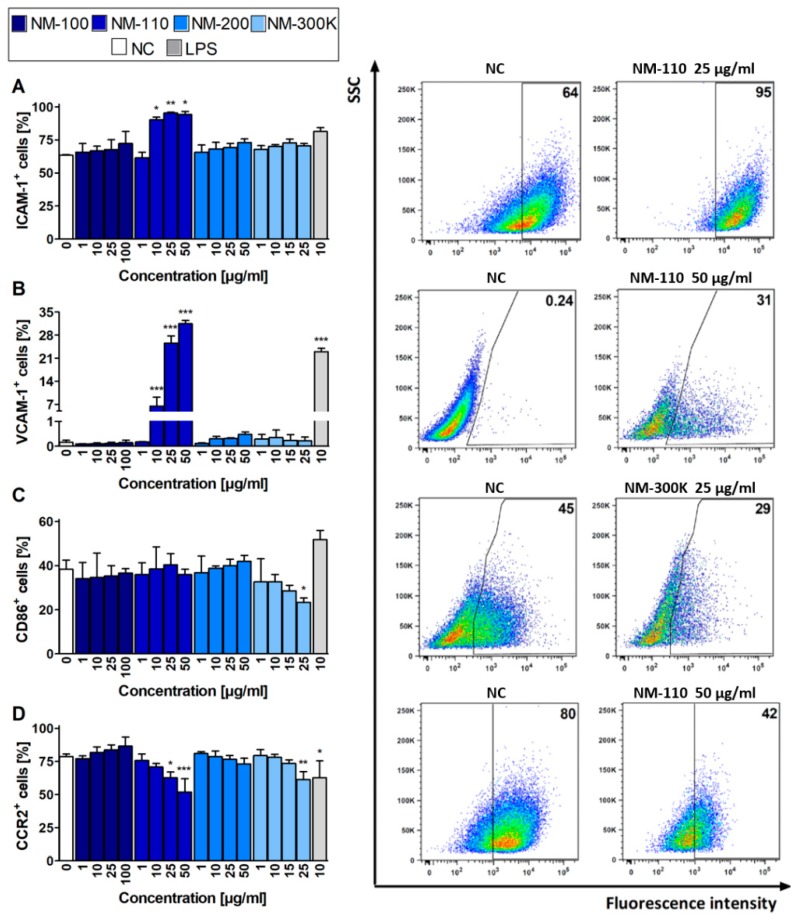
Flow cytometry analyses of expression of selected markers. Bar plots (expressed as mean ± SD) indicate percentages of ICAM-1 (**A**), VCAM-1 (**B**), CD86 (**C**), and CCR2 (**D**) positive cells after 24-h exposure. Representative dot plots for each marker demonstrate differences for selected exposures to NPs compared to untreated cells (NC). Data presented in dot plots are from one experiment and represent selected results of three independent experiments. ANOVA, followed by Dunnett’s Multiple Comparison, was used for statistical analysis of the results. * *p* < 0.05, ** *p* < 0.01, *** *p* < 0.001. SSC—side-scattered light.

**Figure 5 nanomaterials-09-00687-f005:**
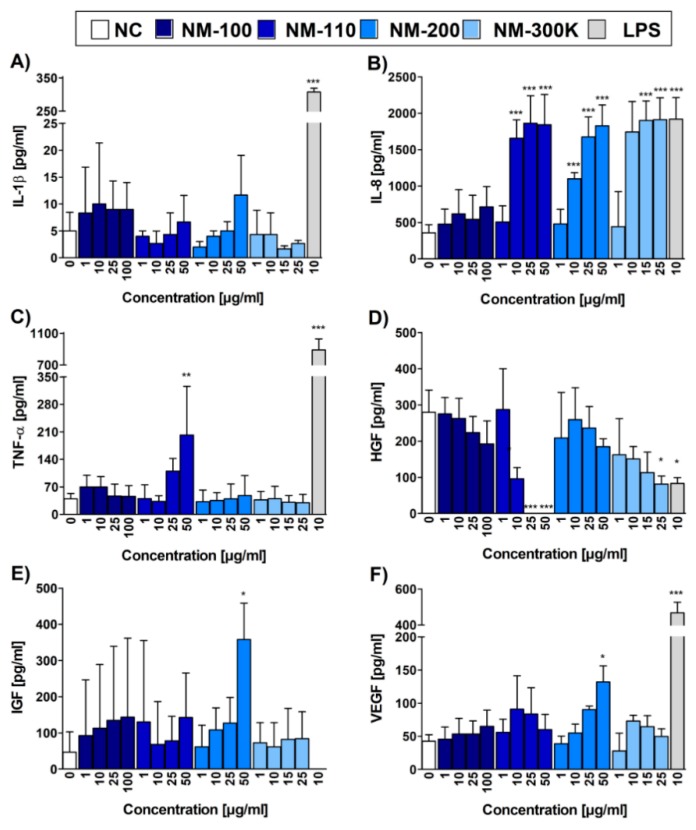
Cytokine, chemokine and growth factor levels evaluated by ELISA. Bar plots represent concentrations of IL-1β (**A**), IL-8 (**B**), TNF-α (**C**), HGF (**D**), IGF-1 (**E**), and VEGF (**F**) in CCM upon 24-h exposure. Three independent experiments were performed. Data are expressed as mean ± SD. Statistical significance of the results was analyzed by ANOVA followed by Dunnett’s Multiple Comparison. * *p* < 0.05, ** *p* < 0.01, *** *p* < 0.001.

**Figure 6 nanomaterials-09-00687-f006:**
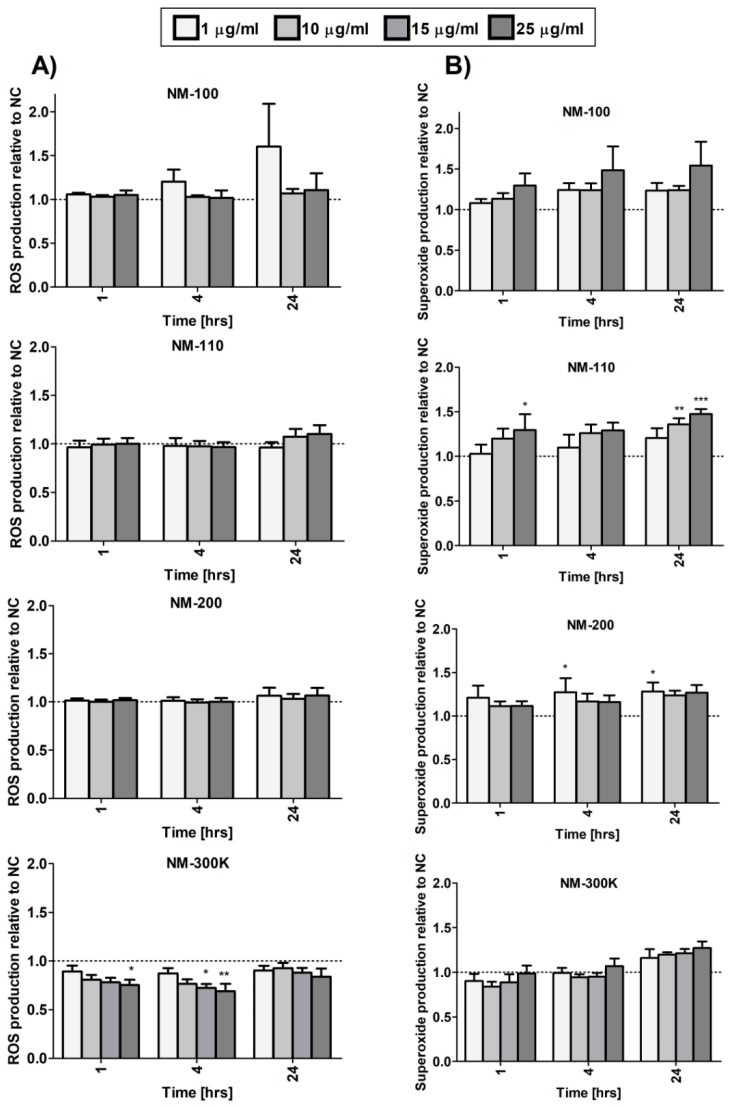
ROS (**A**) and superoxide (**B**) generation in differentiated THP-1 cells after 1-, 4-, and 24-h exposure to NMs measured using a Cellular ROS/Superoxide Detection Assay Kit (Abcam). Results are normalized to negative control values. Statistical significance was analyzed by ANOVA followed by Dunnett’s Multiple Comparison. * *p* < 0.05, ** *p* < 0.01, *** *p* < 0.001.

**Figure 7 nanomaterials-09-00687-f007:**
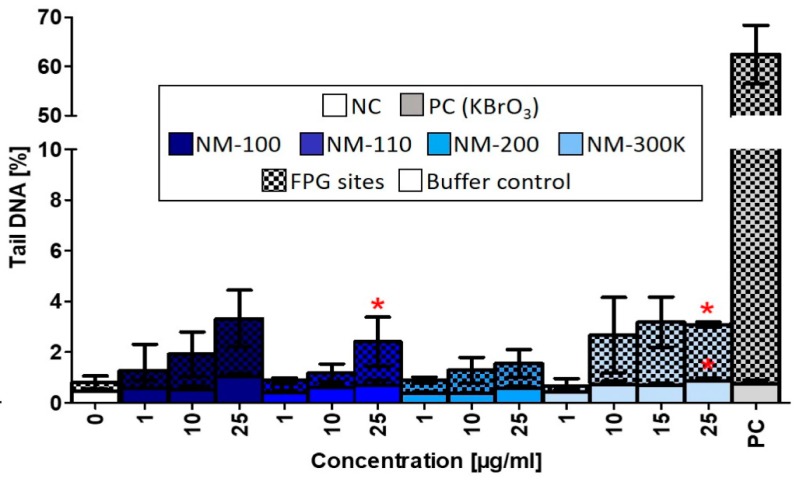
DNA damage evaluated by alkaline comet assay modified with FPG after 24-h exposure to NMs. Buffer controls represent DNA strand breaks. FPG sites corresponding to oxidized purines were obtained by subtracting the % tail DNA of slides incubated with the buffer only from % tail DNA of slides treated with FPG enzyme. Data are reported as the average of the median values of % tail DNA. Statistical significance was analyzed by ANOVA followed by Dunnett’s Multiple Comparison. *p* < 0.05.

**Table 1 nanomaterials-09-00687-t001:** Sequence of primers used in RT-qPCR.

Symbol	Ref Seq ID	Oligonucleotide	
**VCAM**	NM_001078	Sense Primer Antisense Primer	CAGGCTAAGTTACATATTGATGACAT -GAGGAAGGGCTGACCAAGAC
**TNFA**	NM_000594	Sense Primer Antisense Primer	AGGTTCTCTTCCTCTCACATAC -ATCATGCTTTCAGTGCTCATG
**CXCL8**	NM_000584	Sense Primer Antisense Primer	CAGAGACAGCAGAGCACAC -AGCTTGGAAGTCATGTTTACAC
**ICAM1**	NM_000201	Sense Primer Antisense Primer	CCTATGGCAACGACTCCTTC -TCTCCTGGCTCTGGTTCC
**CD86**	NM_175862	Sense Primer Antisense Primer	TTGATAATGGGATGAATGGAAGGA -CGTAGGACATCTGTAGGCTAAG
**CCR2**	NM_00112304	Sense Primer Antisense Primer	CTGAACAGAGAAAGTGGATTGAAC -CTGATAAACCGAGAACGAGATGT
**IL1B**	NM_000576	Sense Primer Antisense Primer	TGGCAATGAGGATGACTTGTTC -CTGTAGTGGTGGTCGGAGATT

**Table 2 nanomaterials-09-00687-t002:** Nanomaterial (NM) characteristics (as provided by the supplier).

	Chemical Composition	Crystallinity (XRD)	Average Diameter ± SD (TEM) [nm]	Shape (Prevailing) (TEM)	Specific Surface (BET) [m^2^/g]	Reference
**NM-100**	TiO_2_	anatase	190 ± 6	spherical/ellipsoidal	10	[13]
**NM-110**	ZnO	zincite	150 (SEM)	hexagonal/cubic	12	[14]
**NM-200**	SiO_2_	amorphous silica	50 ± 51	spherical/ellipsoidal	189	[12]
**NM-300 K**	Ag	metallic silver	17 ± 3	spherical	NA	[11]

NA—data not available; XRD—X-ray diffraction; TEM—transmission electron microscopy; SEM—scanning electron microscopy; BET—Brunauer–Emmett–Teller method.

**Table 3 nanomaterials-09-00687-t003:** Particle size distribution and zeta potential of NP dispersions (NP concentrations were 2.56 mg/mL and 25 µg/mL for dispersion in ddH_2_O-BSA and CCM respectively).

	Hydrodynamic Size (PDI) [nm]			Zeta Potential [mV]		
Sample	ddH_2_O (1 h)	CCM (1 h)	CCM (24 h)	ddH_2_O (1 h)	CCM (1 h)	CCM (24 h)
**NM-100**	256 ± 4 (0.18)	235 ± 1 (0.16)	254 ± 4 (0.33)	NA	−16.0	−16.5
**NM-110**	244 ± 2 (0.13)	144 ± 1 (0.31)	142 ± 3 (0.29)	−12.4	−16.3	−17.4
**NM-200**	207 ± 11 (0.29)	32 ± 17 (0.79)	22 ± 15 (0.52)	−32.9	−15.1	−16.1
**NM-300 K**	77 ± 2 (0.38)	73 ± 13 (0.43)	74 ± 7 (0.49)	−4.8	−16.0	−15.0

NA—data not available; PDI—polydispersity index; CCM—complete cell culture medium.

**Table 4 nanomaterials-09-00687-t004:** Immunologically relevant molecules selected based on microarray analyses in THP-1 macrophages exposed to NPs and their functions.

Molecule	Alternative Name	Functions	Gene
IL-1β	*Lymphocyte activating factor*	*Inflammatory response*	*IL1B*
CD54	Intercellular adhesion molecule 1 (ICAM-1)	Cell–cell interactions, endothelial transmigration	*ICAM1*
TNF-α	-	Proinflammatory cytokine	*TNFA*
CD192	C-C chemokine receptor type 2 (CCR2)	Receptor for CCL2 (monocyte infiltration)	*CD192*
CD106	Vascular adhesion molecule 1 (VCAM-1)	Cell adhesion, leukocyte-endothelial cell signal transduction	*VCAM1*
CD86	B7-2	Costimulatory signals for T cell activation and survival	*CD86*
IL-8	Chemokine (C-X-C motif) ligand 8 (CXCL8)	Chemotaxis, primarily of neutrophils	*CXCL8*

CD—Cluster of Differentiation.

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
