# Peer review of "Molecular Responses in THP-1 Macrophage-Like Cells Exposed to Diverse Nanoparticles"

_nanomaterials, 2019, doi:10.3390/nano9050687_

Round 1
Reviewer 1 Report
The manuscript is very interesting and the authors highlighted the strengt, but also the weakness of the study. About the weakness they said in the Conclusion that the results remain to be confirmed in human primary macrophages. Therefore, my suggestion is to improved the study using monocyte-derived macrophages in vitro obtained from one or two healthy donors.
Other minor comments in the text in attached pdf file.

Author Response
The manuscript is very interesting and the authors highlighted the strengt, but also the weakness of the study. About the weakness they said in the Conclusion that the results remain to be confirmed in human primary macrophages. Therefore, my suggestion is to improved the study using monocyte-derived macrophages in vitro obtained from one or two healthy donors.
Response: We thank the Reviewer for this comment. We are aware that using macrophages obtained from healthy donors would be the best option for interpretation and validity of the results. Before starting our experiments, we considered this option, unfortunately, we did not find a source of a sufficient amount of blood needed for such study. Also, in case cells from more than one human subject are used, genetic variability between donors might affect the results and would thus make drawing conclusions more difficult. Thus, given the circumstances, we assume that application of THP-1 cells was a reasonable compromise for our study. However, the results need to be interpreted with caution. This fact is mentioned in the conclusions of our study.
Other minor comments in the text in attached pdf file.
Response: The changes are highlighted in the revised text.
1. hepatocyte growth factor (HGF) – corrected
2. Figure 3 – the order of the results presentation was changed
3. Figure 4 – the panels inverted
4. Figure 6 – ROS generation – the text was modified
5. Discussion – the first immune cells – the text was modified
Figure 4 – comments: The FACS results showed in the right panel correspond to the molecules showed in the left panel. The right panel serves for illustrative purposes and reports just one concentration for the selected NM. The size of the right panel was adjusted to make the presentation clearer.
Reviewer 2 Report
In this manuscript, Authors have evaluated the response of macrophage-like cells exposed to various nanoparticles.
This manuscript is interesting, and deal with a very interesting topic. The presentation is rational and professional. Overall, I suggest its acceptance.
My only minor suggestion is in the following:
-Authors, that have rationally employed standardized materials from JCR repository, state that NM-110 is not (bio)degradable. On the other hand, it was broadly demonstrated that silica nanomaterials are (bio)degradable (see for example, doi: 10.1002/ppsc.201800464 and related). Maybe these silica nanomaterials are somewhat different, but Authors should discuss this point in their manuscript.
Author Response
Authors, that have rationally employed standardized materials from JCR repository, state that NM-110 is not (bio)degradable. On the other hand, it was broadly demonstrated that silica nanomaterials are (bio)degradable (see for example, doi: 10.1002/ppsc.201800464 and related). Maybe these silica nanomaterials are somewhat different, but Authors should discuss this point in their manuscript.
Response: We added information on possible biodegradability of silica nanoparticles to the Discussion. We also cite the recommended study.
Reviewer 3 Report
This manuscript investigated the molecular responses or change of expression of immunologically genes, when nanoparticles are added to the immune cells. MTS assay, RNA isolation, RT-qPCR, TEM and so on were conducted in this investigation. This work is well organized, well written and thorough. A lot of experiments and measurements were conducted. I only have the following minor comments for further improvement:
The authors should add some statements regarding justification of the nanoparticle selection.
The authors should also justify the concentration of nanoparticle selected in this study, whether it is based on preclinical or clinical model.
What do you mean “macrophage-like”? Why didn’t you use the macrophage cell model?
Figure 1: Can you add one more x- and y-axis to the right and top of the figure? In the subfigure, “NC” or “NS”?
Figure 2: orientations of A vs A’, B vs B’ and C vs C’ are not matched. Also, please label the ruler bars in the subfigures.
Figure 6: Can you provide a color figure like the previous figures?
Can you discuss whether the electron beam from the TEM will affect the immune cell (i.e. the interaction between the electron and nanoparticle)?
I found some typos in this manuscript. It is good to edit it further.
Author Response
This manuscript investigated the molecular responses or change of expression of immunologically genes, when nanoparticles are added to the immune cells. MTS assay, RNA isolation, RT-qPCR, TEM and so on were conducted in this investigation. This work is well organized, well written and thorough. A lot of experiments and measurements were conducted. I only have the following minor comments for further improvement:
The authors should add some statements regarding justification of the nanoparticle selection.
Response: We selected metal- or metal oxide-based nanoparticles with different chemical properties and size. They are commercially used and are included in the OECD priority list. Finally, they are among representative test materials provided by European Commission's JRC Nanomaterials Repository. For these reasons, biological effects induced by exposure to these nanomaterials are of interest, as they may indicate possible negative health impacts in humans. We mentioned the reasons in the text (Discussion, second paragraph).
The authors should also justify the concentration of nanoparticle selected in this study, whether it is based on preclinical or clinical model.
Response: The concentrations of nanoparticles were primarily selected based on our pilot experiments followed by tests of cytotoxicity. For the study, non-cytotoxic concentrations were further used. We added the explanation to the Discussion (second paragraph).
What do you mean “macrophage-like”? Why didn’t you use the macrophage cell model?
Response: “Macrophage-like” THP-1 cells are monocytes that are differentiated into cells that resemble properties of mature macrophages. We did not use macrophage cell model, as there is no suitable human cell line available. RAW 264.7 macrophages are of murine origin and human macrophages would be difficult to obtain in sufficient amount. Before starting our experiments, we considered the option of using human macrophages, unfortunately, we did not find a source of a sufficient amount of blood needed for such study. Also, if cells from more than one human subject are used, genetic variability between donors might affect the results and would thus make drawing conclusions more difficult. Thus, given the circumstances, we assume that application of THP-1 cells was a reasonable compromise for our study. However, the results need to be interpreted with caution. This fact is mentioned in the conclusions of our study.
Figure 1: Can you add one more x- and y-axis to the right and top of the figure? In the subfigure, “NC” or “NS”?
Response: The Figure was corrected. “NS” is correct – non-significant (explained in the figure legend).
Figure 2: orientations of A vs A’, B vs B’ and C vs C’ are not matched. Also, please label the ruler bars in the subfigures.
Response: We are not sure what is meant by not matched orientations of subfigures A vs A’, B vs B’ and C vs C’. The subfigures are organized so that they fit well into a single figure in the landscape orientation. The ruler bars are labeled.
Figure 6: Can you provide a color figure like the previous figures?
Response: Colors in previous figures were used to distinguish between NMs when results from several NMs were presented in a single figure (one color = one NM). However, in Figure 6 the results for individual NMs are presented separately. We therefore believe that currently used presentation should be sufficient for data presentation.
Can you discuss whether the electron beam from the TEM will affect the immune cell (i.e. the interaction between the electron and nanoparticle)?
Response: This possibility is now mentioned in the Results section; a reference has been added.
I found some typos in this manuscript. It is good to edit it further.
Response: The manuscript was checked for typos.